# A Portable Tool for Spectral Analysis of Plant Leaves That Incorporates a Multichannel Detector to Enable Faster Data Capture

Juan Botero-Valencia [1,*] , Erick Reyes-Vera [2] , Elizabeth Ospina-Rojas [1] and Flavio Prieto-Ortiz [3]

1   Grupo Sistemas de Control y Robótica, Faculty of Engineering, Instituto Tecnológico Metropolitano, Medellin 050034, Colombia; elizabethospina@itm.edu.co
2   Grupo de Automática, Electrónica y Ciencias Computacionales, Faculty of Engineering, Instituto Tecnológico Metropolitano, Medellin 050034, Colombia; erickreyes@itm.edu.co
3   Grupo GAUNAL, Faculty of Engineering, Universidad Nacional de Colombia, Bogotá 111321, Colombia; faprietoo@unal.edu.co
*   Correspondence: juanbotero@itm.edu.co

**Abstract:** In this study, a novel system was designed to enhance the efficiency of data acquisition in a portable and compact instrument dedicated to the spectral analysis of various surfaces, including plant leaves, and materials requiring characterization within the 410 to 915 nm range. The proposed system incorporates two nine-band detectors positioned on the top and bottom of the target surface, each equipped with a digitally controllable LED. The detectors are capable of measuring both reflection and transmission properties, depending on the LED configuration. Specifically, when the upper LED is activated, the lower detector operates without its LED, enabling the precise measurement of light transmitted through the sample. The process is reversed in subsequent iterations, facilitating an accurate assessment of reflection and transmission for each side of the target surface. For reliability, the error estimation utilizes a color checker, followed by a multi-layer perceptron (MLP) implementation integrated into the microcontroller unit (MCU) using TinyML technology for real-time refined data acquisition. The system is constructed with 3D-printed components and cost-effective electronics. It also supports USB or Bluetooth communication for data transmission. This innovative detector marks a significant advancement in spectral analysis, particularly for plant research, offering the potential for disease detection and nutritional deficiency assessment.

**Keywords:** low-cost fabrication; multichannel detector; multilayer perceptron (MLP); plant leaves; spectral analysis

## 1. Introduction

Optical spectrometers that operate in the visible and infrared regions have become indispensable tools in our daily lives, fundamentally transforming our comprehension and interaction with a wide range of materials and substances, regardless of their physical state. This technology is extensively employed in several areas, including agriculture, energy, chemistry, materials science, and food production [1–9]. The operating principle of visible–short-wave near-infrared (VIS-NIR-SWIR) spectroscopy is the transfer of energy between light and matter. This fundamental principle allows for the identification and study of different compounds by examining their distinct spectral properties. In fact, the spectral characteristics within the VIS-NIR-SWIR range are intricately linked to the vibrational modes of the functional groups found in the target substance [2,10]. Indeed, the vibrational modes serve as a distinctive mark, resembling a fingerprint, which enables scientists and researchers to obtain valuable information about the composition and characteristics of the substances being studied. Moreover, the non-invasive, high-resolution, and non-destructive aspects of Vis-SWNIR spectroscopy are very beneficial in plant studies [3,11,12]. It enables

quick assessments without causing damage to the samples, making it ideal for the real-time monitoring of plant health and nutrient levels.

Several studies used spectral analysis to investigate plants in the past [12–20]. For example, Ge et al. explored the utility of VIS-NIR-SWIR as a high-throughput instrument for measuring six leaf parameters of maize plants: chlorophyll content (CHL), leaf water content (LWC), specific leaf area (SLA), nitrogen (N), phosphorus (P), and potassium (K) [12]. To estimate leaf attributes from hyperspectral data, two multivariate modeling techniques, namely, partial least-squares regression (PLSR) and support vector regression (SVR), were used to calculate several vegetation indices. The results show that the proposed methodologies can be used to predict CHL levels but not the other leaf metrics. In 2020, Xiong et al. used Vis-SWNIR spectroscopy and chemometric techniques to investigate the potassium concentration in fresh lettuce [21]. The authors demonstrated how PLS and SVR can be used to assess potassium concentrations in these plants. The results show that the PLS model outperformed the SVR model in terms of prediction. In 2021, Mahajan and collaborators characterized the foliar nutrient status of mango through the development of spectral indices, multivariate calculus, chemometrics, and machine learning (ML) models [15]. Recently, Wang et al. created an inexpensive spectrometer for assessing the quality attributes of tea tree leaves [13]. The device utilizes the random forest method to accurately predict the quantities of nitrogen, chlorophyll, and free amino acids. However, the predictions for moisture, polyphenol, and sugar exhibit some inconsistency, which impacts the overall accuracy.

Conventional high-performance spectrometers have historically depended on robust and costly setups that integrate bulky dispersive elements, long optical paths, and complex mechanical processes. Additionally, expert personnel are required to carry out the tests in some cases. However, the demand for versatile spectroscopy tools tailored to specific applications has driven innovation in the form of more compact, cost-effective designs and user-friendly systems. These modern spectrometers prioritize portability, affordability, durability, and energy efficiency, making them suitable for a diverse range of scenarios [4,9,22–26]. The rising demand for mini- and micro-spectrometers across different industries has resulted in a projected market value of almost USD 900 million, showing a significant surge in interest in this technology [27]. Coronel-Reyes et al. created an inexpensive NIR spectrometer to evaluate the duration of egg storage [7]. Consequently, the eggs were subjected to spectral analysis by utilizing reflectance measurements ranging from 740 to 1070 nm. In addition, appropriate predictive models were constructed using PLS and artificial neural network (ANN) regression approaches, which aided in determining the freshness of eggs. In 2020, Laganovska and her colleagues presented a cost-effective, self-contained portable spectrophotometer [24]. In addition, this device offers exceptional performance and achieves a resolution of 15 nm. In 2021, Botero-Valencia et al. introduced a cost-effective and wireless IoT multispectral acquisition device aimed at improving the availability of spectrum data for diverse applications [23]. This device utilizes the functionalities of small-scale spectrometers and Internet of things (IoT) technologies, hence creating possibilities for more extensive spectrum investigations in various domains. Another low-cost spectrometer was developed for biochemical assays [28], measuring the milk quality [4], and the testing of citrus cultivars [1]. These devices typically utilize artificial intelligence approaches to overcome the restrictions imposed by the detectors and other inexpensive components. Nevertheless, these algorithms necessitate substantial computational resources, resulting in increased energy consumption. Consequently, these devices are unsuitable for integration into Internet of things (IoT) networks. Due to this argument, there has been a significant inclination toward adopting a range of TinyML techniques, which is a specialized area within machine learning that aims to facilitate the integration of ML applications into small, energy-efficient, cost-effective devices [29–31]. Likewise, TinyML allows for the analysis and interpretation of data directly on devices, enabling immediate decision-making and action. Furthermore, several of these systems do not account for the measurement of multispectral data on both sides of the leaf. This

is crucial considering the varied manifestations of diseases and the need to assess their transmittance simultaneously.

In this work, we present a cutting-edge device that effortlessly integrates all of the aforementioned unique characteristics. The design features two AS7341 detectors strategically placed at the sample's upper and lower interfaces. Each detector has an independently programmable light-emitting diode (LED) system that allows for the fine control of the incident light spectrum. Importantly, depending on the precise LED activation sequence, these detectors can concurrently evaluate the reflectance and transmittance characteristics of the sample under test. In the proposed device, when the upper LED module is turned on, the lower detector goes into a passive state, allowing for the evaluation of light transmission properties across the sample. Likewise, the detecting mechanism's reliability was validated. An in-depth error approximation strategy was used for this purpose, employing the standard color checker technique. The system was then subjected to data alignment using a multilayer perceptron (MLP) algorithm, ensuring the best possible correlation of the acquired data with predetermined reference standards.

This calibration process in particular is fully integrated with the dedicated microcontroller unit (MCU) via cutting-edge TinyML technology, enabling accurate and reliable real-time acquiring of spectral data. Furthermore, the use of TinyML allows for the proposed spectrometer to consume less power than other previously presented options. Finally, all of these components are assembled in a 3D-printed chassis that is meticulously constructed, which reduces the cost of the proposed device.

## 2. Materials and Methods

### 2.1. Multispectral Sensor

The AS7341 sensor (Adafruit, New York, NY, USA) is a versatile and high-performance 11-channel multispectral sensor designed for accurate color detection and extensive spectral analysis applications [32]. It effectively spans a broad spectral range from 350 nm to 1000 nm, encompassing the visible spectrum, near-infrared (NIR), and clear light. With eight optical channels dedicated to the visible spectrum, one for NIR light, and an additional one for clear light, it also includes a specialized channel for detecting 50 Hz or 60 Hz ambient light flicker. Employing nano-optic deposited interference filter technology, this sensor seamlessly integrates filters into standard CMOS silicon, enabling a streamlined and cost-effective sensor design. Equipped with a built-in aperture, it efficiently regulates the influx of light into the sensor array, with control accessible through a serial I²C interface. It is available in an ultra-low-profile package, featuring dimensions of 3.1 mm × 2 mm × 1 mm, making it an optimal choice for space-constrained engineering applications. In the case described in this study, two AS7341 sensors were employed to evaluate the reflection of each side of the plant leaf while simultaneously assessing the transmission. Each sensor is paired with a digitally controllable LED, namely, the EAHC2835WD6, which further streamlines the acquisition process. Figure 1b shows the relative luminous intensity for the EAHC2835WD6 LED, which was taken with an OHSP-350C (Ocean Optics, Orlando, FL, USA). The spectrum depicted illustrates the LED's effectiveness across the wavelength range of 400 to 750 nm. Figure 1a illustrates an approximation of the spectral distribution of the bands (relative sensitivity) in the multispectral sensor. The bandwidths (BWs) of the channels range from approximately 26 to 90 nm. The datasheet does not specify the BW of the 910 nm channel. The AS7341 is a versatile sensor that allows both the gain and the integration time to be configured via software, which allows the sensitivity of the sensor to be adjusted within the program. In the case of the data used in this work, a gain of 32 was used.

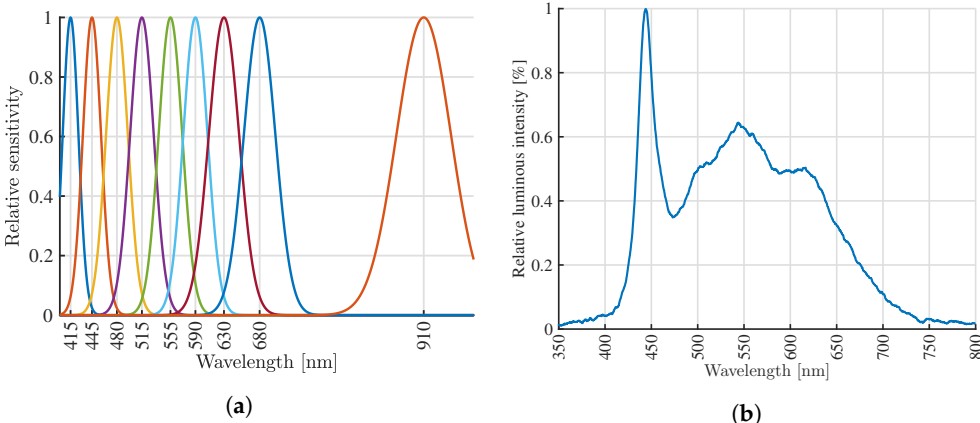

(**a**)

(**b**)

**Figure 1.** Relative sensitivity and luminous in AS7341 integrated board. (**a**) Relative sensitivity of the AS7341. (**b**) Relative luminous intensity profile of the EAHC2835WD6 LED.

### 2.2. Mechanical Design

The entire structure of the proposed spectrometer was 3D-printed using polylactic acid (PLA) material. The first components, namely, MCU_cover and MCU_support (parts 1 and 2), were specifically designed to encase the microcontroller and battery, with MCU_support featuring an aperture for the microcontroller display. An additional component, namely, MUX_support (part 3), is used to hold and cover the multiplexer and for toggling the activation between the two spectrometers. Inside the device, two AS734 multispectral sensors are employed and are positioned opposite each other to evaluate the transmittance and reflectance of the leaf specimen placed between them. The first sensor is shielded by components 4 and 5, denoted as top_base and top_cover, respectively, while the second sensor is protected by components 6 and 7, named bottom_cover and bottom_base. This design allows the system to be opened and closed, thereby enabling the adjustment of the separation between the spectrometers for non-invasive leaf measurements. Finally, components 8 and 9, identified as top_hinge and bottom_hinge, function as pivotal elements, facilitating the system's open-and-close movements. Figure 2 illustrates the configuration of the proposed spectrometer, together with a detailed inventory of the components comprising the proposed design. Additionally, the table includes a hyperlink to the online repository, where the files are accessible in STL format for printing and utilization by interested users.

### 2.3. Electronic Design

Figure 3 shows the electronic wiring diagram of the components used in the presented system. To accommodate the requirement of utilizing two AS7341 sensors, which only support I²C communication and cannot be modified, it became imperative to employ the DFR0576 digital 1-to-8 I²C multiplexer. We utilized two of its outputs for this purpose. This is connected to the I²C bus of the microcontroller, namely, the LILYGO-T-HMI-ESP32-S3. Furthermore, the system is equipped with a 450 mAh lithium battery, enabling it to function independently in the field.

### 2.4. Color Checker

Calibration is an essential stage in the development of a low-cost spectrometer. Currently, there are various approaches available for this purpose, depending on the specific application. In our particular scenario, a color checker, as stated in reference [33] was used. A color checker is a tool used in spectroscopy and photometry to standardize and assess the precision of a spectrum [33–35]. The color checker seen in Figure 4 consists of a sequence of color patches that possess predetermined and standardized reflectance values. As a result, these patches exhibit distinct and consistent colors that are precisely determined in terms of their wavelength and reflectance intensity. To provide a visual representation, we overlapped each spectrum in the box with the corresponding color patch. This demon-

strates how the spectrum was modified based on the studied color. Figure 4 depicts a color checker with 24 unique colors [33], which was referenced as a "patch" during the analysis. Within each patch, and simply as a representation, the x-axis indicates the wavelength between 400 and 980 nm, and the y-axis shows the normalized reflectivity. The suggested spectrometer operates in the visible and near-infrared (NIR) wavelength range. The color checker then indicates each color's reflectance across a wavelength range of 400 nm to 1000 nm. The color checker for black showed a reflectance value of 0.0125, as expected, whereas the color checker for white showed a reflectance value of 0.8812.

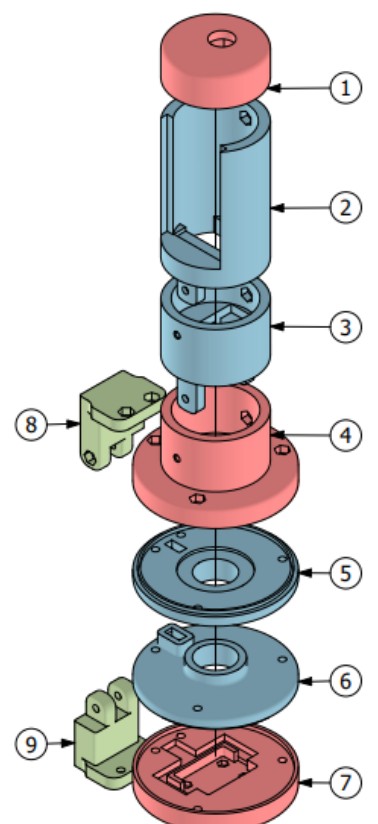

| Name | Link to file |
|---|---|
| 01_MCU_cover.STL | https://osf.io/3dbnm |
| 02_MCU_support.STL | https://osf.io/b6fsm |
| 03_MUX_support.STL | https://osf.io/k6bs3 |
| 04_Top_base.STL | https://osf.io/7bpfj |
| 05_Top_cover.STL | https://osf.io/wxgsq |
| 06_Bottom_cover.STL | https://osf.io/f2639 |
| 07_Bottom_base.STL | https://osf.io/x3zsw |
| 08_Top_hinge.STL | https://osf.io/7crwd |
| 09_Bottom_hinge.STL | https://osf.io/6u2n3 |

**Figure 2.** Assembly and list of mechanical parts.

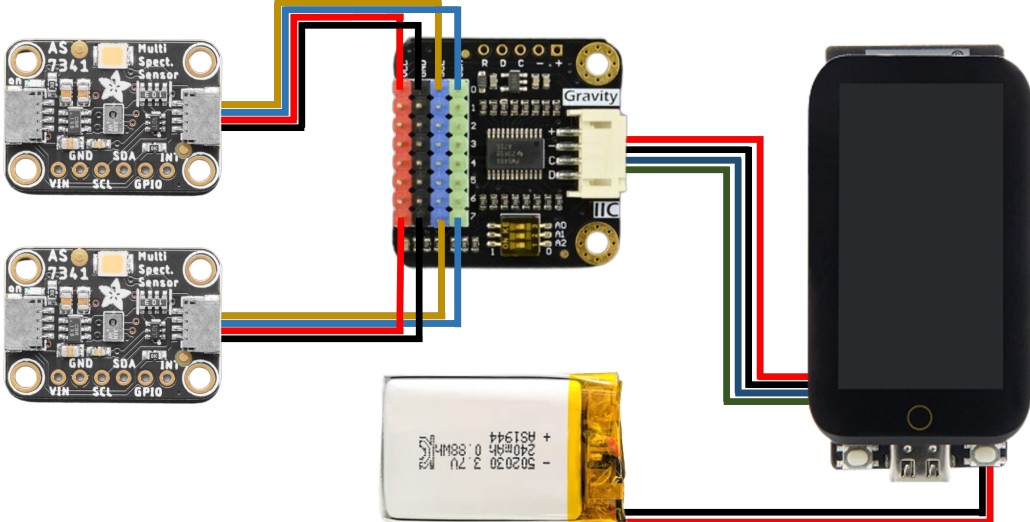

**Figure 3.** Electronic connection diagram.

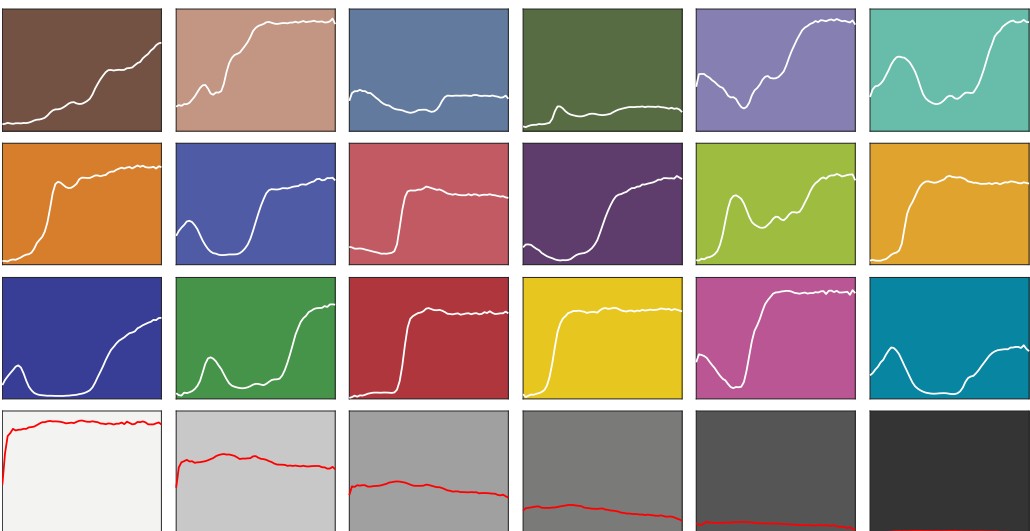

**Figure 4.** Color checker and reflectivity curves [33]. In the figure, the distribution of the rows and columns corresponds to the original Color checker, and within each patch the reflectance curve is shown as a reference.

*2.5. Machine Learning Algorithm*

The data provided by the AS7341 sensor is in a raw format expressed in counts. To calibrate these values to a standardized reflectivity measure, the color checker mentioned in Section 2.4 was employed to capture the values with each sensor. Subsequently, these values were adjusted using a supervised machine learning method, specifically, a multilayer perceptron (MLP) [36]. The MLP method was selected due to its ability to describe intricate interactions between inputs and outputs, making it particularly advantageous in spectral analysis cases where the relationships may exhibit nonlinear behavior. Furthermore, the MLP approach possesses the capacity to extrapolate from a restricted amount of training data [37–39]. In this particular instance, the MLP model comprises nine neurons in the input layer, two hidden layers, and nine neurons in the output layer, as can be seen in Figure 5. Every neuron in the input layer receives reflectance data that are measured at nine distinct wavelengths: 415, 445, 480, 515, 555, 590, 630, 680, and 910 nm. These same wavelengths correspond to the expected outputs of the model. The neurons in each layer are completely interconnected with the neurons in the subsequent layer, facilitating the unidirectional flow of information. The network architecture is specifically built to enable the mapping of input data, which represents the spectrum reflectance measurements, to the matching output data, which are the reflectance values at specified wavelengths. The MLP model, which is designed for feedforward operation, adeptly captures intricate connections between the input and output data, facilitating accurate spectrum analysis and predictions. On the other hand, the weights of these connections are modified during the training process. Each neuron's output is multiplied by the connection weight, then undergoes a rectified linear unit (ReLU) activation function and is summed with the outputs of other linked neurons. In this case, a ReLU function was selected because of its reduced training time and straightforward integration into embedded devices.

Tiny machine learning (TinyML) is a revolutionary branch of artificial intelligence that enables the execution of ML models on low-power devices, like MCUs. This breakthrough technology empowers the implementation of machine learning models for sensor data analysis directly on the device, resulting in lower power consumption and feasible deployment on battery-powered devices. The benefits of TinyML are manifold: local data processing minimizes the latency, enhancing the efficiency and expediting decision-making without the need for information transfer to a server. Moreover, reduced power consumption is critical for battery-constrained devices, while local data storage heightens the security by mitigating risks associated with information transfer. The implementation process typically commences with training the model on a higher-power computer using TensorFlow, fol-

lowed by optimization with TensorFlow Lite to reduce the size and complexity. The model is then adapted to the MCU's capabilities, and necessary code is written to load and run the model on the device, undergoing tests and performance adjustments as needed.

In this work, the perceptron training process was carried out using the TensorFlow library. The main objective of this implementation was the subsequent integration of the model in an embedded system to have the data adjusted in real-time. This optimization process is achieved through the adaptation of the model to the tiny machine learning framework, which allows for its efficient implementation on an MCU with limited resources.

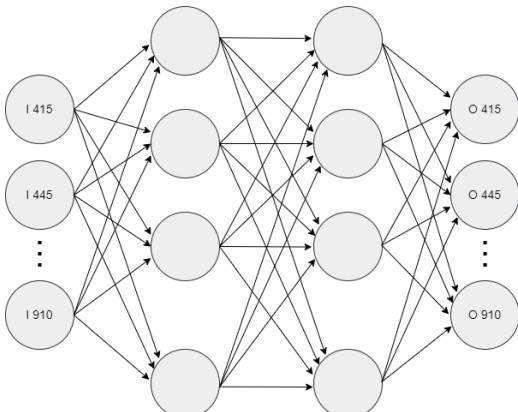

**Figure 5.** Architecture of the proposed MLP.

### 2.6. Measurement of Reflectance Using an Optical Spectrum Analyzer (OSA)

The experimental setup illustrated in Figure 6 was employed to obtain the reflectance spectrum of colored paper sheets with different colors. This setup employed a laser-controlled plasma-type white light source (Energetiq EQ-99-FC, Wilmington, MA, USA) that emitted between 190 and 2500 nm, and an optical spectrum analyzer (OSA) (Yokogawa, AQ6373, Tokyo, Japan). Both instruments in this scenario were connected using a fiber optic probe (Ocean Optics, QR200-7-UV-VIS), which enabled the measurement of the reflectance of the sample being analyzed. As depicted in Figure 6, the probe was strategically positioned at a 45° angle to mitigate the impact of undesirable reflections that might compromise the measurement accuracy. This orientation served the dual purpose of not only minimizing unwanted reflections but also preventing incident light from reflecting directly back toward the light source. To ensure that the measurement was always taken in the same position, i.e., at the same distance between the probe and the sample, a holder was employed. This holder was allowed for securing the position of the probe with a screw.

The OSA measurements were conducted using a linear scale with a precision of 5 nm, covering a wavelength range of 350 nm to 980 nm with increments of 0.31 nm. The initial phase of the experiment entailed capturing diffuse reflectance spectra spanning from 350 nm to 980 nm. This was achieved by utilizing a certified reflectance standard, specifically the USRS-99-010 model from Labsphere (Hewlett Packard, Palo Alto, CA, USA). This accessory helped to calibrate the white color, thereby guaranteeing the consistency and accuracy of measurements. Its function was to emulate an ideal target with nearly perfect reflectance, facilitating precise and reliable data collection throughout the experiment. Through spectral analysis of this reference sample, we established a standard against which the reflectance characteristics of other materials could be evaluated. This comparison was crucial to ensure precise and uniform measurements across all samples. By employing a reference sample, we could compensate for variations in the measurement configuration, such as fluctuations in the light intensity or sensor sensitivity, ensuring the successful normalization of the collected data. Subsequently, a series of colored paper sheet samples, with each one displaying a distinct color, were subjected to experimentation utilizing the aforementioned setup to acquire the reflectance spectrum for each colored paper sample.

The selected colors were red (C01), orange (C02), yellow (C03), fuchsia (C04), violet (C05), dark blue (C06), light blue (C07), dark green (C08), and light green (C09).

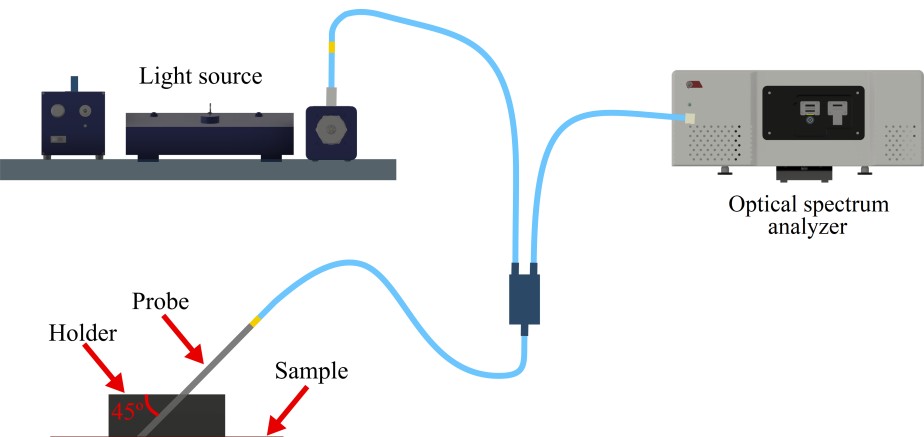

**Figure 6.** Experimental setup used to measure the reflectance with different colors using an OSA.

### 3. Results and Discussions

#### 3.1. Assembly and Manufacturing

First, each element of the suggested affordable spectrometer was constructed utilizing the 3D-printing technique, specifically employing the Creality Ender 5 printer. Although polylactic acid (PLA) is a suitable material for this purpose, acrylonitrile styrene acrylate (ASA) is advised due to its superior mechanical strength and ability to protect against UV radiation. Subsequently, the assembly of all components was executed following the instructions outlined in Section 2.2, following the assembly of the mechanical components. M3 screws and safety nuts were utilized to ensure secure assembly and prevent misalignment during operation. Additionally, two 30 mm watch glasses can be used to hide the sensors for added protection against dust ingress and facilitate cleaning. The mechanical design has space to accommodate them. The electronic system described in Section 2.3 was implemented in a manner that ensured optimal sensor placement to prevent any errors. An essential aspect was to ensure that each component was properly fixed to prevent any movement, as this could lead to measurement inaccuracies. Therefore, it was crucial to align the size of the mechanical components with the specific shape of each electronic element. Conversely, the software responsible for acquiring, storing, and processing data was installed in the MCU. As stated in Section 2.5, this software incorporates a TinyML model.

Figure 7a shows a picture of the entire view of the proposed spectrometer, and each component was sequentially labeled according to the diagram presented in Figure 2 to facilitate their identification. Figure 7b displays an image depicting the specific region of the device where the samples to be analyzed are positioned. The construction and integration of an affordable spectrometer was accomplished, demonstrating its user-friendly nature and suitability for use in external laboratory settings.

#### 3.2. Model Selection and Training Error

As previously stated, the calibration dataset was drawn from the data acquired using the color checker [33], which consisted of 24 patches with known reflectance curves, and the raw data taken from each of the sensors, considering the potential variances that may exist between them. Table 1 presents a summary of the training for the upper sensor. A total of 16 different MLPs were trained, varying the number of layers and the number of neurons in each of the hidden layers. Given the limited amount of data, each neural network was trained five times, and the error was averaged to ensure stability. The metrics used were the mean absolute error (MAE) and total P. The total P refers to the total number of parameters in the model, in this case, the parameters were the weights and biases used

to connect the neurons. The number of parameters is important as it affects the size and complexity of the model. A model with many parameters can be more accurate but may also be more challenging to train and might require more training data. In our case, it would also imply more complexity in deploying it on the embedded system. According to the findings presented in Table 1, the model that utilized two hidden layers with 64 neurons in each layer was chosen since the inclusion of a third layer did not result in a substantial enhancement in performance. Conversely, the inclusion of a third hidden layer resulted in an almost double increase in the value of the total P metric compared with the scenario when only two hidden layers were employed. Therefore, the model would become more complex and robust, thereby increasing the difficulty of its integration into the embedded system. Furthermore, it is evident that the inclusion of a fourth hidden layer negatively impacted the outcomes, which was a phenomenon that might be attributed to limited available data.

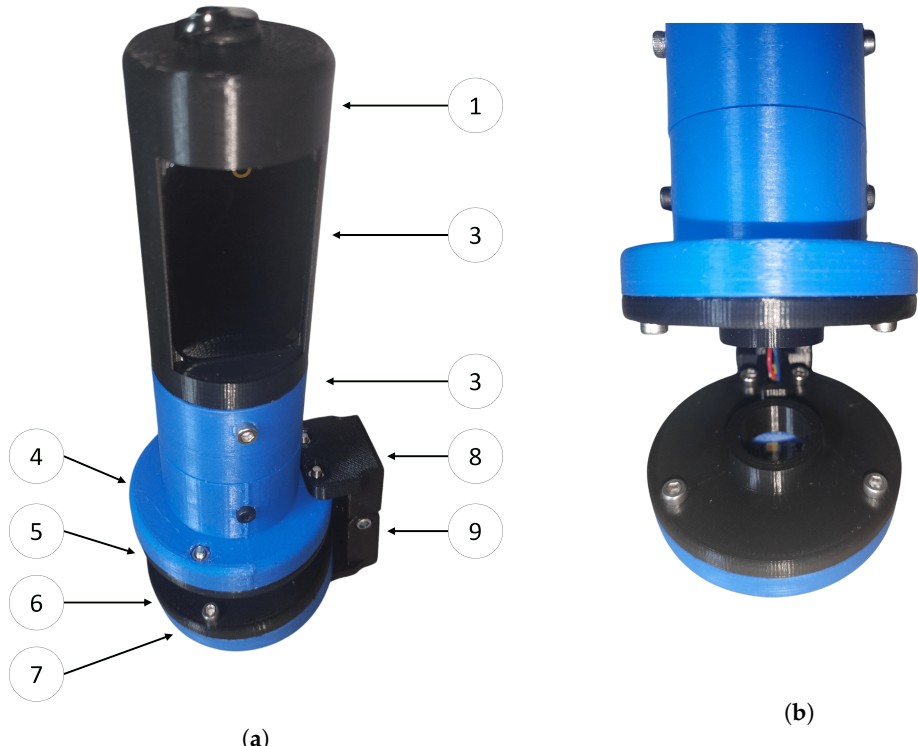

(**a**)

(**b**)

**Figure 7.** Photograph of the constructed spectrometer. (**a**) View of the entire device. (**b**) Detailed view of the area where the samples were located. The numbers in the figure correspond to the identification in the Figure 2.

All the conducted training sessions involved a rigorous execution process spanning 1000 epochs, each with a modest batch size of 3, while ensuring data diversity through the strategic utilization of the "shuffle" option. In the neural network architecture, only the bias was implemented in the initial layer, while the loss function was carefully designated as the mean absolute error (MAE), indicating a robust approach to error measurement. Notably, these intensive training operations were seamlessly orchestrated within the user-friendly Google Colab platform and were consistently completed within a highly efficient time frame, with no instance surpassing the 200 s mark. Both tools used have an open-source license that allows users to utilize, modify, and redistribute the software without any restrictions.

A comparison analysis was conducted to assess the performance of the suggested spectrometer and the implemented ML model. In the initial scenario, the measurements were compared by utilizing the color checker specified in [33] as the reference technique. There, the authors report the reflectance curves of 24 different patches. All these reference

patches were then experimentally characterized using the proposed low-cost spectrometer. Figure 8 shows only six of them for clarity (patch 01, patch 04, patch 10, patch 14, patch 15, and patch 19). A comparison of the reference reflectance (MEA), raw reflectance (SEN), and adjusted reflectance (ADJ) is shown in each scenario. The ideal MLP configuration (64 neurons in each of the two hidden layers) was used to obtain the ADJ findings.

**Table 1.** Training error analysis of the ML model used. The MAE and total P metrics were analyzed as functions of the hidden layers and the number of neurons per layer. The best results have been highlighted in the table to facilitate their identification.

| Hidden Layers | Metric | Neurons per Layer | | | |
|---|---|---|---|---|---|
| | | 8 | 16 | 32 | 64 |
| 1 | MAE | 0.1315 | 0.0929 | 0.0721 | 0.0645 |
| | Total P | 152 | 304 | 608 | 1216 |
| 2 | MAE | 0.1203 | 0.0874 | 0.0562 | 0.0398 |
| | Total P | 216 | 560 | 1632 | 5312 |
| 3 | MAE | 0.1469 | 0.0639 | 0.0581 | 0.0356 |
| | Total P | 280 | 816 | 2656 | 9408 |
| 4 | MAE | 0.0862 | 0.0538 | 0.0521 | 0.0515 |
| | Total P | 344 | 1072 | 3680 | 13,504 |

Based on the obtained results, it is evident that there was a strong correlation between the MEA and SEN measurements, particularly at shorter wavelengths. Additionally, there were slight discrepancies observed at longer wavelengths. However, these disparities were rectified when employing the ideal machine learning model, hence significantly improving the accuracy of the obtained results. In fact, the methodology utilized successfully achieved a strong correlation between the ADJ results and the reference reflectance values across the whole research range (400 to 980 nm).

On the other hand, a quantitative study of the training error was conducted on the 24 instances. The findings are succinctly presented in Table 2. This table displays the training error analysis, where SEN represents the absolute error against the raw measurement and ADJ denotes the error with respect to the adjusted values. To conduct a thorough investigation, the MAE analysis was computed for each patch and each channel. The obtained results corroborate that the error was significantly reduced in all cases, even at a wavelength of 910 nm, which was the channel that first exhibited the highest level of inaccuracy. Finally, the overall MAE was calculated. The implemented MLP successfully reduced the obtained overall MAE from 0.2979 to 0.0398. In other words, the proposed model reduced the training error by a factor of at least three.

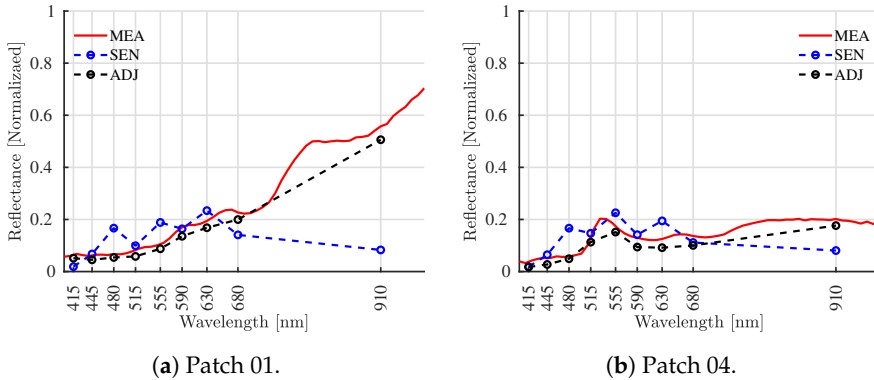

(**a**) Patch 01.　　　　　　　　　　　　　(**b**) Patch 04.

**Figure 8.** *Cont.*

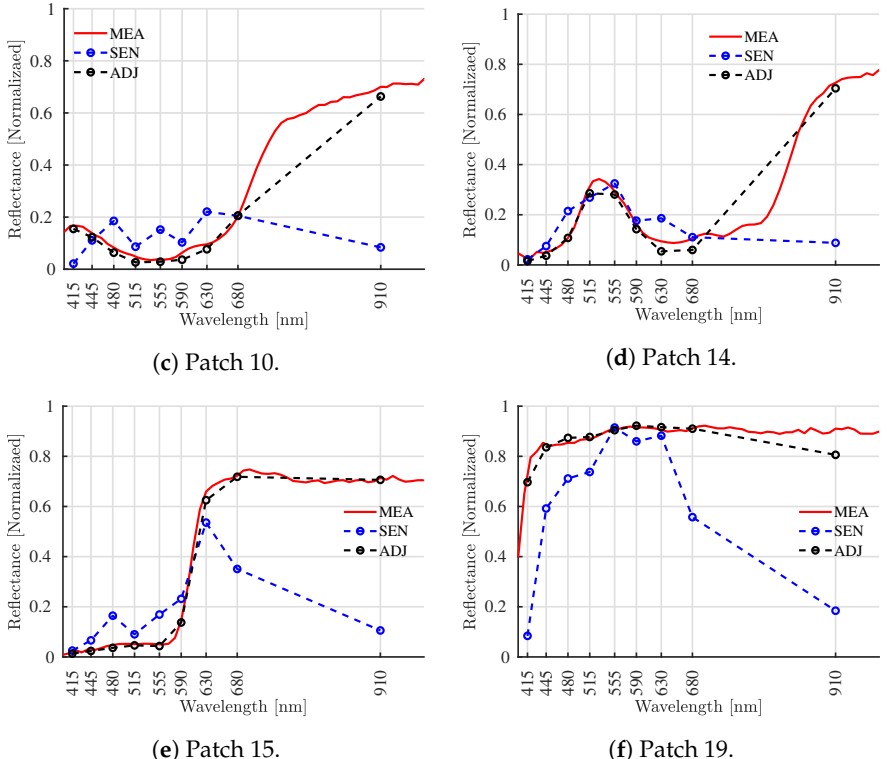

**Figure 8.** Comparison of the fits of multiple patches. MEA is the color checker reference reflectance. SEN is the raw reflectance, while ADJ is the best MLP setup-adjusted reflectance.

**Table 2.** Comparative training error while employing the proposed spectrometer for characterizing the patches reported in [33].

| Patch | 415 | 445 | 480 | 515 | 555 | 590 | 630 | 680 | 910 | MAE |
|---|---|---|---|---|---|---|---|---|---|---|
| **P01 [SEN]** | 0.1012 | 0.0118 | 0.2234 | 0.0368 | 0.1864 | 0.0130 | 0.0884 | 0.1934 | 1.0643 | 0.2132 |
| **P01 [ADJ]** | 0.0279 | 0.0385 | 0.0290 | 0.0551 | 0.0401 | 0.0770 | 0.0589 | 0.0614 | 0.1156 | 0.0559 |
| **P02 [SEN]** | 0.3808 | 0.1078 | 0.0103 | 0.1876 | 0.1047 | 0.0943 | 0.0738 | 0.7812 | 1.7260 | 0.3852 |
| **P02 [ADJ]** | 0.0313 | 0.0029 | 0.0150 | 0.0357 | 0.0036 | 0.0210 | 0.0463 | 0.0104 | 0.0646 | 0.0257 |
| **P03 [SEN]** | 0.6498 | 0.2385 | 0.0053 | 0.0768 | 0.1175 | 0.0029 | 0.1751 | 0.0848 | 0.4497 | 0.2000 |
| **P03 [ADJ]** | 0.0403 | 0.0250 | 0.0161 | 0.0721 | 0.0065 | 0.0913 | 0.0531 | 0.0571 | 0.0225 | 0.0427 |
| **P03 [SEN]** | 0.0429 | 0.0284 | 0.2440 | 0.0331 | 0.1150 | 0.0263 | 0.1560 | 0.0562 | 0.2717 | 0.1082 |
| **P04 [ADJ]** | 0.0449 | 0.0554 | 0.0190 | 0.0431 | 0.0510 | 0.0801 | 0.0747 | 0.0808 | 0.0567 | 0.0562 |
| **P04 [SEN]** | 0.9657 | 0.3412 | 0.0406 | 0.1565 | 0.0973 | 0.0891 | 0.0339 | 0.4455 | 1.8225 | 0.4436 |
| **P05 [ADJ]** | 0.0342 | 0.0177 | 0.0080 | 0.0426 | 0.0296 | 0.0665 | 0.0592 | 0.0616 | 0.0724 | 0.0435 |
| **P05 [SEN]** | 0.7042 | 0.2666 | 0.2109 | 0.3292 | 0.1404 | 0.1496 | 0.1012 | 0.1640 | 1.7104 | 0.4196 |
| **P06 [ADJ]** | 0.0497 | 0.0378 | 0.0101 | 0.0236 | 0.0211 | 0.1040 | 0.0426 | 0.0310 | 0.0197 | 0.0377 |
| **P07 [SEN]** | 0.0105 | 0.0773 | 0.2417 | 0.0857 | 0.2464 | 0.1702 | 0.1416 | 0.6489 | 1.5542 | 0.3529 |
| **P07 [ADJ]** | 0.0050 | 0.0303 | 0.0031 | 0.0267 | 0.0228 | 0.0100 | 0.0364 | 0.0184 | 0.0912 | 0.0271 |
| **P08 [SEN]** | 0.5888 | 0.1922 | 0.0655 | 0.0156 | 0.2270 | 0.0996 | 0.2344 | 0.1208 | 1.3312 | 0.3195 |
| **P08 [ADJ]** | 0.0204 | 0.0534 | 0.0349 | 0.0416 | 0.0097 | 0.0933 | 0.0382 | 0.0942 | 0.0209 | 0.0452 |
| **P09 [SEN]** | 0.2375 | 0.0238 | 0.2259 | 0.0750 | 0.2776 | 0.1239 | 0.0531 | 0.5936 | 1.0351 | 0.2939 |
| **P09 [ADJ]** | 0.0256 | 0.0132 | 0.0354 | 0.0044 | 0.0226 | 0.0151 | 0.0561 | 0.0554 | 0.0406 | 0.0298 |
| **P10 [SEN]** | 0.3255 | 0.0625 | 0.2291 | 0.0849 | 0.2573 | 0.0887 | 0.2812 | 0.0069 | 1.3823 | 0.3021 |
| **P10 [ADJ]** | 0.0263 | 0.0351 | 0.0439 | 0.0497 | 0.0188 | 0.0606 | 0.0419 | 0.0087 | 0.0832 | 0.0409 |

**Table 2.** *Cont.*

| Patch | 415 | 445 | 480 | 515 | 555 | 590 | 630 | 680 | 910 | MAE |
|---|---|---|---|---|---|---|---|---|---|---|
| **P11 [SEN]** | 0.0169 | 0.0674 | 0.2801 | 0.0546 | 0.0208 | 0.1054 | 0.1356 | 0.2398 | 1.3895 | 0.2567 |
| **P11 [ADJ]** | 0.0119 | 0.0022 | 0.0069 | 0.0271 | 0.0058 | 0.0449 | 0.0694 | 0.0259 | 0.0369 | 0.0257 |
| **P12 [SEN]** | 0.0000 | 0.1064 | 0.2430 | 0.1812 | 0.0394 | 0.2092 | 0.1318 | 0.7150 | 1.2315 | 0.3175 |
| **P12 [ADJ]** | 0.0238 | 0.0170 | 0.0311 | 0.0092 | 0.0269 | 0.0396 | 0.0538 | 0.0048 | 0.0462 | 0.0280 |
| **P13 [SEN]** | 0.3315 | 0.0997 | 0.1245 | 0.1055 | 0.2678 | 0.1416 | 0.2724 | 0.1534 | 1.1325 | 0.2921 |
| **P13 [ADJ]** | 0.0025 | 0.0553 | 0.0270 | 0.0426 | 0.0185 | 0.0575 | 0.0544 | 0.0562 | 0.1125 | 0.0474 |
| **P14 [SEN]** | 0.0158 | 0.0537 | 0.2410 | 0.0965 | 0.0725 | 0.0297 | 0.2062 | 0.0123 | 1.4320 | 0.2400 |
| **P14 [ADJ]** | 0.0307 | 0.0325 | 0.0007 | 0.0576 | 0.0269 | 0.0461 | 0.0894 | 0.1021 | 0.0494 | 0.0484 |
| **P15 [SEN]** | 0.0098 | 0.0846 | 0.2627 | 0.0875 | 0.2666 | 0.1928 | 0.2773 | 0.8341 | 1.3656 | 0.3757 |
| **P15 [ADJ]** | 0.0169 | 0.0103 | 0.0251 | 0.0116 | 0.0156 | 0.0189 | 0.0771 | 0.0095 | 0.0184 | 0.0226 |
| **P16 [SEN]** | 0.0557 | 0.1246 | 0.3437 | 0.0742 | 0.0513 | 0.1108 | 0.0020 | 0.5942 | 1.3167 | 0.2970 |
| **P16 [ADJ]** | 0.0057 | 0.0023 | 0.0045 | 0.0040 | 0.0098 | 0.0355 | 0.0323 | 0.0089 | 0.1889 | 0.0324 |
| **P17 [SEN]** | 0.7284 | 0.2087 | 0.1461 | 0.0436 | 0.2874 | 0.1565 | 0.1141 | 0.8319 | 1.7266 | 0.4715 |
| **P17 [ADJ]** | 0.0184 | 0.0290 | 0.0576 | 0.0576 | 0.0087 | 0.0505 | 0.0622 | 0.0544 | 0.0079 | 0.0385 |
| **P18 [SEN]** | 0.4446 | 0.1416 | 0.0936 | 0.1539 | 0.1740 | 0.1279 | 0.2828 | 0.1200 | 0.7277 | 0.2518 |
| **P18 [ADJ]** | 0.0471 | 0.0310 | 0.0173 | 0.0716 | 0.0111 | 0.1187 | 0.0802 | 0.1067 | 0.0570 | 0.0601 |
| **P19 [SEN]** | 1.4357 | 0.5724 | 0.3193 | 0.2977 | 0.0149 | 0.1225 | 0.0530 | 0.7924 | 1.6289 | 0.5819 |
| **P19 [ADJ]** | 0.0611 | 0.0254 | 0.0440 | 0.0154 | 0.0083 | 0.0174 | 0.0234 | 0.0012 | 0.2344 | 0.0478 |
| **P20 [SEN]** | 1.1232 | 0.4129 | 0.1501 | 0.2183 | 0.0188 | 0.1730 | 0.0699 | 0.5621 | 0.9140 | 0.4047 |
| **P20 [ADJ]** | 0.1143 | 0.0207 | 0.0133 | 0.0011 | 0.0197 | 0.0025 | 0.0375 | 0.0301 | 0.1775 | 0.0463 |
| **P21 [SEN]** | 0.7640 | 0.2806 | 0.0368 | 0.1537 | 0.0237 | 0.1206 | 0.0007 | 0.3400 | 0.4858 | 0.2451 |
| **P21 [ADJ]** | 0.0419 | 0.0096 | 0.0003 | 0.0010 | 0.0198 | 0.0264 | 0.0475 | 0.0143 | 0.0400 | 0.0223 |
| **P22 [SEN]** | 0.4042 | 0.1519 | 0.0880 | 0.0614 | 0.0959 | 0.0415 | 0.0905 | 0.1240 | 0.1185 | 0.1307 |
| **P22 [ADJ]** | 0.0077 | 0.0453 | 0.0262 | 0.0447 | 0.0356 | 0.0908 | 0.0660 | 0.0655 | 0.0887 | 0.0523 |
| **P23 [SEN]** | 0.1137 | 0.0146 | 0.2116 | 0.0432 | 0.1964 | 0.0685 | 0.1903 | 0.0288 | 0.0338 | 0.1001 |
| **P23 [ADJ]** | 0.0116 | 0.0417 | 0.0096 | 0.0541 | 0.0080 | 0.0639 | 0.0822 | 0.1089 | 0.1339 | 0.0571 |
| **P24 [SEN]** | 0.0241 | 0.0847 | 0.2855 | 0.0995 | 0.2447 | 0.1177 | 0.2359 | 0.1029 | 0.1286 | 0.1471 |
| **P24 [ADJ]** | 0.0042 | 0.0117 | 0.0029 | 0.0355 | 0.0096 | 0.0398 | 0.0370 | 0.0378 | 0.0165 | 0.0217 |
| **MAE [SEN]** | 0.3948 | 0.1564 | 0.1801 | 0.1147 | 0.1477 | 0.1073 | 0.1417 | 0.3561 | 1.0825 | 0.2979 |
| **MAE [ADJ]** | 0.0293 | 0.0268 | 0.0200 | 0.0345 | 0.0188 | 0.0530 | 0.0550 | 0.0461 | 0.0748 | 0.0398 |

*3.3. Validation Error*

A second test was carried out to validate the proposed device. For this particular instance, a total of nine colored paper sheets, each of a distinct color, were examined. The reference method employed was the setup outlined in Section 2.6, wherein measurements were conducted with the Yokogawa AQ6373 optical spectrum analyzer, which is a sophisticated instrument known for its high resolution.

Figure 9 demonstrates the outcomes achieved by utilizing only six out of the nine colored paper sheets, specifically the red, yellow, light-blue, light-green, violet, and dark-green sheets. As in the previous case study, MEA refers to the reference reflectance obtained with the OSA. The raw reflectance is denoted as SEN, and ADJ represents the adjusted reflectance obtained using the best MLP setup.

As in the prior case, the spectrometer provided the correct results, especially at shorter wavelengths. Furthermore, the proposed correction model greatly improved the findings by lowering the difference between the reference reflectance and the obtained reflectance. The results show that the MLP-based model was well-suited for these circumstances, as it allowed for adjustments even when dealing with highly nonlinear reflectance behavior,

which is fundamentally complex. For example, while evaluating the red paper sheet (see Figure 9a), reflectance values of more than 0.8 were found for wavelengths greater than 630 nm. Similarly, when inspecting the violet paper sheet, there was a fall in reflectance between 445 nm and 630 nm (see Figure 9c), indicating substantial absorption within this range, which is consistent with prior studies.

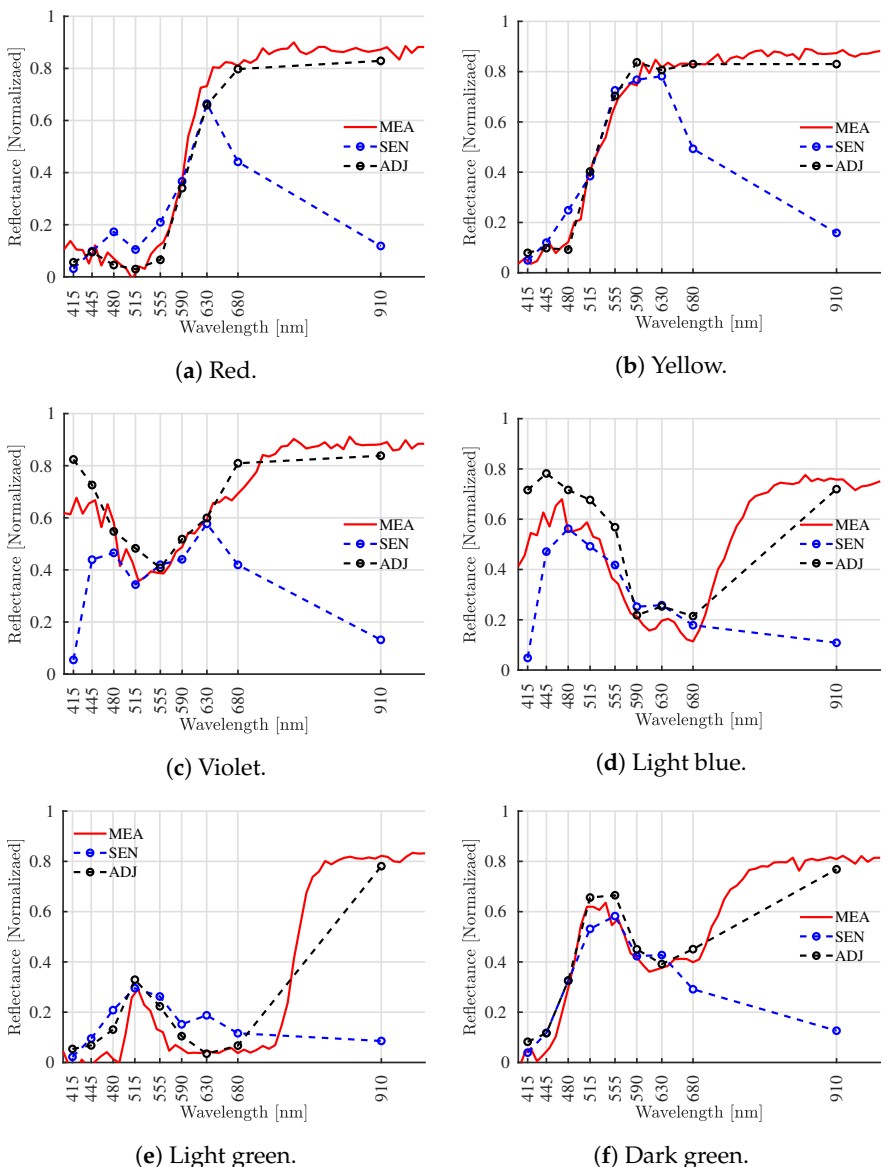

**Figure 9.** Validation with a reference method based on the use of a Yokogawa AQ6373 optical spectrum analyzer. MEA is the OSA reference reflectance. SEN is the raw reflectance, while ADJ is the best MLP setup-adjusted reflectance.

Table 3 shows the detailed error for each band and each color analyzed with the OSA. Once again, SEN refers to the raw sensor data, and ADJ refers to the data adjusted with the selected MLP. It can be observed that the MAE in this case decreased from 0.1137 to 0.03901 and that the error after adjustment remained close to that of validation, indicating that there was no significant overfitting during the training. Again, it can be observed that the error before adjustment (SEN) was higher in the 910 nm band, but afterward, the error significantly decreased, and the highest error in the adjusted values (ADJ) was observed in the 415 nm band. It can also be seen that the highest error of the adjusted data was found in C07, corresponding to light blue, with an MAE value of 0.0829.

**Table 3.** Comparative validation error while employing the proposed spectrometer for characterizing colored paper sheets. The reference method employed in this instance was the Yokogawa AQ6373 optical spectrum analyzer.

| Color | 415 | 445 | 480 | 515 | 555 | 590 | 630 | 680 | 910 | MAE |
|---|---|---|---|---|---|---|---|---|---|---|
| **C01 [SEN]** | 0.0610 | 0.0084 | 0.0698 | 0.0605 | 0.0579 | 0.0015 | 0.0461 | 0.2506 | 0.5129 | 0.1187 |
| **C01 [ADJ]** | 0.0443 | 0.0062 | 0.0168 | 0.0093 | 0.0400 | 0.0193 | 0.0494 | 0.0084 | 0.0297 | 0.0248 |
| **C02 [SEN]** | 0.0386 | 0.0210 | 0.0976 | 0.0704 | 0.0651 | 0.0066 | 0.0053 | 0.1931 | 0.4643 | 0.1069 |
| **C02 [ADJ]** | 0.0043 | 0.0430 | 0.0008 | 0.0740 | 0.0079 | 0.0773 | 0.0232 | 0.0728 | 0.0281 | 0.0368 |
| **C03 [SEN]** | 0.0022 | 0.0108 | 0.0864 | 0.0128 | 0.0430 | 0.0152 | 0.0230 | 0.2283 | 0.4864 | 0.1009 |
| **C03 [ADJ]** | 0.0226 | 0.0041 | 0.0202 | 0.0002 | 0.0280 | 0.0624 | 0.0062 | 0.0008 | 0.0297 | 0.0193 |
| **C04 [SEN]** | 0.1804 | 0.0722 | 0.0006 | 0.0054 | 0.0486 | 0.0056 | 0.0561 | 0.2513 | 0.5210 | 0.1268 |
| **C04 [ADJ]** | 0.0447 | 0.0042 | 0.0589 | 0.0106 | 0.0303 | 0.0014 | 0.1478 | 0.0316 | 0.0305 | 0.0400 |
| **C05 [SEN]** | 0.4016 | 0.1510 | 0.0793 | 0.0342 | 0.0219 | 0.0321 | 0.0118 | 0.1860 | 0.5104 | 0.1587 |
| **C05 [ADJ]** | 0.1216 | 0.0437 | 0.0233 | 0.0605 | 0.0137 | 0.0207 | 0.0036 | 0.0792 | 0.0300 | 0.0440 |
| **C06 [SEN]** | 0.1365 | 0.0200 | 0.0457 | 0.0268 | 0.0792 | 0.0604 | 0.0654 | 0.0024 | 0.4364 | 0.0970 |
| **C06 [ADJ]** | 0.0119 | 0.0717 | 0.0308 | 0.0321 | 0.0207 | 0.0362 | 0.0151 | 0.0251 | 0.0247 | 0.0298 |
| **C07 [SEN]** | 0.3073 | 0.0869 | 0.0080 | 0.0454 | 0.0432 | 0.0265 | 0.0418 | 0.0438 | 0.4413 | 0.1160 |
| **C07 [ADJ]** | 0.1468 | 0.1245 | 0.1129 | 0.0800 | 0.1455 | 0.0035 | 0.0385 | 0.0684 | 0.0258 | 0.0829 |
| **C08 [SEN]** | 0.0315 | 0.0702 | 0.1317 | 0.0139 | 0.0925 | 0.0664 | 0.0964 | 0.0536 | 0.5008 | 0.1174 |
| **C08 [ADJ]** | 0.0531 | 0.0515 | 0.0791 | 0.0369 | 0.0658 | 0.0343 | 0.0075 | 0.0201 | 0.0279 | 0.0418 |
| **C09 [SEN]** | 0.0045 | 0.0501 | 0.0204 | 0.0599 | 0.0164 | 0.0038 | 0.0347 | 0.0731 | 0.4641 | 0.0808 |
| **C09 [ADJ]** | 0.0249 | 0.0494 | 0.0210 | 0.0250 | 0.0728 | 0.0224 | 0.0104 | 0.0353 | 0.0275 | 0.0321 |
| **MAE [SEN]** | 0.1293 | 0.0545 | 0.0599 | 0.0366 | 0.0520 | 0.0242 | 0.0423 | 0.1425 | 0.4820 | 0.1137 |
| **MAE [ADJ]** | 0.0527 | 0.0442 | 0.0404 | 0.0365 | 0.0472 | 0.0308 | 0.0335 | 0.0380 | 0.0282 | 0.0391 |

Finally, the proposed spectrometer was employed to analyze and evaluate the optical properties of a plant leaf. Through this practical application, we were able to verify the efficiency of the system in real-life situations, showcasing its value in the spectral analysis of plants, encompassing both reflectance and transmittance. Figure 10a displays the measured transmittance (TRA) and reflectance (REF) of the sun-exposed side of a leaf, while Figure 10b illustrates the same parameters for the shaded side of the leaf, as obtained using the suggested affordable spectrometer. As expected, the reflectance of the sun-exposed side was reduced compared with the shaded side of the leaf. This reduction was attributable to the leaf's optimization for absorbing sunlight efficiently, thereby enhancing its photosynthetic activity. In addition, the drop in the reflectance spectrum curve at 555 nm of a leaf could be attributed to the absorption characteristics of chlorophyll. Chlorophyll, which is the pigment responsible for photosynthesis in plants, absorbs light most efficiently in the blue (427–476 nm) and red (618–780 nm) regions of the electromagnetic spectrum, with minimal absorption in the green region. This phenomenon is known as the "green gap" or "chlorophyll absorption dip". On the other hand, a considerable increase in reflectance in the infrared band was also expected since plant leaves absorb infrared radiation less efficiently. Infrared radiation is related to heat, and an increase in the reflectance in this region aids in preventing the leaf from overheating.

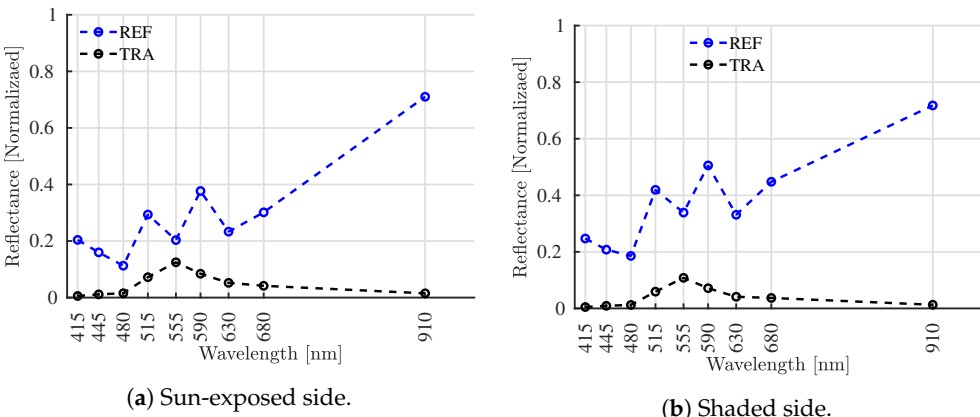

**Figure 10.** Measurements of transmittance and reflectance on a vegetable leaf.

## 4. Conclusions

First, a cost-effective and user-friendly optical spectrometer was designed, fabricated, and experimentally validated. This spectrometer employs a training technique based on a comparison with a color checker, which greatly enhances its accuracy and applicability. The training results demonstrated that the machine learning model utilizing a MLP model effectively decreased the training error, even when working with a limited amount of data. Furthermore, the efficacy of the suggested spectrometer in precisely assessing the spectrum of materials exhibiting different colors in the visible and NIR regions was verified by a comparative technique utilizing a high-resolution optical spectrum analyzer. The validation findings indicate that the proposed spectrometer accurately characterized the spectrum properties, and the MAE could be minimized using ML models.

In addition, the integration of a tiny machine learning model within the MCU allowed for real-time data processing and reduced power consumption, enhancing the efficiency and usability of the device. This innovation opens up possibilities for further developments in the field of portable spectroscopy. On the other hand, the proposed device's capabilities enabled its use in leaf characterization, showcasing its proficiency in analyzing the spectral attributes of leaves in both reflection and transmission. This is particularly advantageous, as certain plants have distinct characteristics or colors on each side of their leaves.

Finally, our research demonstrated the feasibility of creating an optical spectrometer that is not only cost-effective and precise but also user-friendly, with the flexibility to be tailored for diverse applications. The novel training and machine learning techniques we introduced, coupled with the incorporation of real-time learning models, present exciting opportunities for further exploration and advancement in this field. This innovation represents a valuable asset that can be seamlessly integrated into agricultural practices, particularly for the ongoing monitoring of plant health.

**Author Contributions:** Conceptualization, J.B.-V. and E.R.-V.; software, J.B.-V. and E.O.-R.; visualization, J.B.-V., E.R.-V. and E.O.-R.; methodology, J.B.-V. and E.R.-V.; validation, E.R.-V. and E.O.-R.; original draft preparation, J.B.-V., E.R.-V., E.O.-R. and F.P.-O.; writing—review and editing, J.B.-V., E.R.-V., E.O.-R. and F.P.-O.; funding acquisition, J.B.-V., E.R.-V., E.O.-R. and F.P.-O. All authors read and agreed to the published version of the manuscript.

**Funding:** This  study was suported by the research project "Diseño, desarrollo y validación de un modelo de detección temprana de Tizón Tardío en cultivos de papa Diacol Capiro mediante análisis de imágenes espectrales adquiridas en los departamentos de Boyacá y Cundinamarca", with the code RC1013-2021 and belonging to the "890-2020 Convocatoria para el fortalecimiento de CTeI en instituciones de educación superior públicas—MINCIENCIAS".

**Informed Consent Statement:** Not applicable.

**Data Availability Statement:** The data presented in this study are available upon request from the corresponding author.

**Acknowledgments:** This study was supported by the Sistemas de Control y Robótica (GSCR) Research Group COL0123701 at the Sistemas de Control y Robótica Laboratory, which is attached to the Instituto Tecnológico Metropolitano.

**Conflicts of Interest:** The authors declare no conflicts of interest. The funders had no role in the study's design; in the collection, analyses, or interpretation of data; in the writing of the manuscript; or in the decision to publish the results.

## Abbreviations

The following abbreviations are used in this manuscript:

| | |
|---|---|
| LED | Light-emitting diode |
| MLP | Multilayer perceptron |
| MCU | Microcontroller unit |
| VIS-NIR-SWIR | Visible–short-wave near-infrared |
| LWC | Leaf water content |
| SLA | Specific leaf area |
| CHL | Chlorophyll content |
| PLSR | Partial least-squares regression |
| SVR | Support vector regression |
| ML | Machine learning |
| ANN | Artificial neural network |
| IoT | Internet of things |
| NIR | Near-infrared |
| CMOS | Complementary metal-oxide semiconductor |
| BW | Bandwidths |
| ReLU | Rectified linear unit |
| OSA | Optical spectrum analyzer |
| PLA | Polylactic acid |
| ASA | Acrylonitrile styrene acrylate |
| UV | Ultraviolet |
| MAE | Mean absolute error |
| MEA | Reference reflectance |
| SEN | Raw reflectance |
| ADJ | Adjusted reflectance |

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
