# Peer review of "A Portable Tool for Spectral Analysis of Plant Leaves That Incorporates a Multichannel Detector to Enable Faster Data Capture"

_instruments, doi:10.3390/instruments8010024_

Round 1
Reviewer 1 Report
Comments and Suggestions for Authors
Dear Authors,
Thank you for preparing quite interesting content. I like the concept of your device and its low-cost build. However, I still have some questions:
1. Paragraph 2.1. Multi-spectral sensor
Did you perform the characterization of the sensors you used? You are presenting only the relative sensitivity of the AS7341 (Figure 1.) The manufacturer's datasheet shows that different channels have different irradiance responses and do not have fixed central wavelengths. The exemplary measured spectral response of the sensor shows a higher sensitivity in the orange-red region and a drop in NIR. There is also changeable gain. What gain was used during the measurements?
In your paper, there is information that the sensor is paired with controllable LED EAHC2835WD6. This LED is not operating in NIR (works in the spectral range 415-750nm) for all possible emission types: warm white, neutral white, and cool white) - could you be so kind and address the issue? It would be appreciated if the spectral characteristics of all used LEDs were performed and shown in your paper.
2. Paragraph 2.4. Color checker
The use of a color checker as a reference is a good idea. It is commonly used in photometry, so Figure 4. is not necessary in my opinion. Maybe I have not understood the explanation you provided.
3. Paragraph 2.6. Measurements of...
Do I understand correctly that the "sheets" you refer to in this section are the color checker fields?
Why the probe is situated at a 45-degree angle? Does it mimic the sensor positioning or there is another explanation?
What is the reference sample that has been used in the measurement? You only mentioned 9 colors (lines 224-226).
4. Paragraph 3.2. Model selection and... & Paragraph 3.3. Validation...
Lines 288-289 you again refer to the color checker? Am I right?
The MEA line was measured with OSA and the SEN and ADJ lines originate from your sensor. Am I right? If so, what was the light source used in the measurements? Was it the same for OSA and your sensor or was it different? If it was different could you compare the raw data from your sensor for two types of illumination? Color checker spectral response will change concerning the illumination.
The problem with the results for the 910nm channel may be due to the lack of illumination if you are using the LED described in Paragraph 2.1.
Comments on the Quality of English Language
The language of the manuscript is decent.
Author Response
Dear Reviewer,
We greatly appreciate your valuable feedback and insightful observations. Your input has played a crucial role in refining several key aspects of our work, ensuring its accuracy and quality.
Attached, you will find a detailed response addressing each of your comments and suggestions. Additionally, we have highlighted the specific changes made in the manuscript to provide clarity and transparency, facilitating the evaluation process.
Thank you once again for your time and valuable contribution to our project.
Warm regards,

Reviewer 2 Report
Comments and Suggestions for Authors
Thanks for submitting this manuscript. It is a very nice work. Here are a few comments that aim to improve the quality of your work that which in general is very good.
1) Give the LED full term at the beginning of the document
2)You use Visible–short–wave near-infrared as Vis–SWNIR but later you use VIS-NIR-SWIR. The second term is more common (or Vis-NIR-SWIR)
3) Line 81 you define machine learning as ML, you have done this before
4)Is AS7341 a market ready product? please add references
5)At the standardization part, how did you check the effects of temperature?
6) At the standardization part, did you perform any linearity correction?
Thank you for your time and good luck in your work!
Author Response

(The authors gave the same response as above.)

Reviewer 3 Report
Comments and Suggestions for Authors
This paper introduced an interesting portable multichannel detector for spectral analysis of plant leaves. Focuses on error estimation utilized a ColorChecker, followed by a Multi-Layer Perceptron (MLP) implementation integrated into the microcontroller unit (MCU).
Questions:
1) What is “tinyML technology for real-time refined data acquisition” mentioned in Abstract. If the authors can give more details of this technology, Including the algorithm implementation technology in the MCU ,the article will have more reference value for researchers.
2) The article says “The increased reflectivity in the electromagnetic spectrum bands at a wavelength of 550 nm corresponds to the green portion of the visible spectrum.……”. But in Fig.10, the reflectance curve actually drops at 550 nm and the transmittance curve peaks at 550nm. Why?
On the whole, I think this work is difficult and meaningful in engineering. But it is more like a general introduction to the composition and operation effect of a new instrument, lacking some technical details.
Author Response

(The authors gave the same response as above.)

Round 2
Reviewer 1 Report
Comments and Suggestions for Authors
Thank you for adressing my comments.
Reviewer 2 Report
Comments and Suggestions for Authors
Thanks for addressing the comments. I hope that my recommendations have helped you. Good luck!